# Cross-Modal Reconstruction for Tactile Signal in Human–Robot Interaction

**DOI:** 10.3390/s22176517

**Published:** 2022-08-29

**Authors:** Mingkai Chen, Yu Xie

**Affiliations:** Key Laboratory of Broadband Wireless Communication and Sensor Network Technology, Ministry of Education, School of Communication and Information Engineering, Nanjing University of Posts and Telecommunications, Nanjing 210003, China

**Keywords:** CNN, cross-modal signal processing, attention mechanism, force estimation

## Abstract

A human can infer the magnitude of interaction force solely based on visual information because of prior knowledge in human–robot interaction (HRI). A method of reconstructing tactile information through cross-modal signal processing is proposed in this paper. In our method, visual information is added as an auxiliary source to tactile information. In this case, the receiver is only able to determine the tactile interaction force from the visual information provided. In our method, we first process groups of pictures (GOPs) and treat them as the input. Secondly, we use the low-rank foreground-based attention mechanism (LAM) to detect regions of interest (ROIs). Finally, we propose a linear regression convolutional neural network (LRCNN) to infer contact force in video frames. The experimental results show that our cross-modal reconstruction is indeed feasible. Furthermore, compared to other work, our method is able to reduce the complexity of the network and improve the material identification accuracy.

## 1. Introduction

With the continuous development of 6G communication, human–robot interaction (HRI) has attracted more and more attention from various industries. From a business point of view, it can improve internet products and benefit enterprises. From the technological development perspective, HRI can facilitate the development of a variety of technologies, including software, hardware, and even neuroscience. Since 2013, with the advent of depth cameras, gesture interaction has been used more and more in HRI. Currently, the main application types include 3D modeling, auxiliary applications, data authentication, navigation, and touch-free control [1]. With the help of 6G communication technology, people will be able to interact with distant robots more quickly as we move forward.

It is equally important to consider tactile information when performing HRI, whether it is material identification or mutual contact force estimation. For example, an experienced surgeon can perform a precise medical diagnosis although a patient is on the other side of the world [2]. Robots can replace humans to explore unknown areas and provide feedback for every feeling [3]. Therefore, a robot should recognize the grasped object and feed the tactile information back to humans in order to achieve better operation and grasping [4]. From what was mentioned above, it is well known that tactile signals play a significant role in HRI.

However, current communication technology cannot meet the requirements of high reliability and low delay in haptic communication, which makes the interaction inadequately smooth. The effect on tactile enhancement primarily remains in the vibration and collision simulation stage. It is difficult to evaluate the interactive experience and efficiency optimization brought by haptic recognition technology. Various improved tactile sensors are also being actively developed, but when the tactile sensing technology matures, the golden age of tactile control will be missed, and so on. Therefore, utilizing the combination of visual information and tactile information can improve the efficiency of human interaction. Therefore, this paper considers the reconstruction of haptic information assisted by video from the perspective of cross-modal signal processing to reduce the difficulty of haptic transmission.

Many studies have been carried out on tactile restoration methods in HRI, especially through the reconstruction of tactile force through visual information. In the process of HRI, the interaction force comprises friction force and contact force. We will explain this in detail in Section 2.

In addition to inferring interaction forces directly from apparent contact images, Tu-Hoa Pham et al. [5] estimated multi-touch-point interaction forces by observing the contact between human hands and objects. For the contact point occlusion case, Kiana Ehsani et al. [6] estimated the contact force of the contact point by restoring the motion of the object through 3D simulation. For fast-moving objects, traditional RGB images may not be able to capture changes. Fariborz Baghaei Naeini et al. [7] used event-based camera images, which reflected the magnitude of contact force through light and dark changes, to estimate contact force.

From the research on the reconstruction of tactile signals, we can get the following characteristics of tactile signals:1.Multidimensional.2.Interactive.3.Contextually relevant.

Because of these features of tactile signaling, in terms of visual information inference, we can determine the following challenges. Most of the existing inference schemes take a single RGB image or other difficult-to-obtain image types as the network input, and the processing requires networks other than a CNN as auxiliaries. In this way, the reconstruction of tactile information from visual information confronts the following challenges:Challenge 1: Hard-to-use video information for reconstruction of haptic information.Challenge 2: The network structure is too complicated and the amount of computation is large.

As a solution to these issues, we propose an LRCNN with LAM. The network uses GOPs as input and sends the video frames to the network after processing. Through iterative training, the network ultimately outputs the predicted contact force. This work makes the following contributions:1.An LRCNN that makes full use of the correlation between GOPs to make the prediction results more accurate is proposed. The network only utilizes the CNN, which reduces the complexity of the network and lowers the requirements for equipment.2.We add the LAM to the LRCNN to make it perform better. Through the LAM, the network can better extract the features of the ROIs so that the results are better. In the case of complex illumination, the LRCNN with LAM improves the performance of the experimental materials beyond 40%. Moreover, the proposed attention mechanism can be better used in combination.

The rest of the paper is organized as follows: In Section 2, related work on video and tactile information processing is presented. The details of the proposed network architecture and attention mechanisms are presented in Section 3. In Section 4, we introduce the database used and give the experimental results and performance evaluations. Finally, we present our conclusions and scope for further research in Section 5.

## 2. Related Work

### 2.1. Video Stream Processing

With advanced camera technology that is integrated with intelligent video analytics (IVA), researchers are becoming increasingly interested in utilizing video signals. In spite of these technological advancements, videos have become increasingly complex and large, which, in turn, requires new algorithms to efficiently and effectively process massive video data. So, before employing visual information, we need to process the acquired visual information to get the information needed for reconstruction. Usually, the background is removed [8], and the desired foreground is extracted, which is divided into three steps: (1) background initialization, (2) foreground detection, and (3) background maintenance.

Background initialization based on low-rank subspace learning can be obtained by learning low-rank subspace. Using local optical flow information, Gutchess et al. [9] proposed a new background initialization algorithm that finds the most likely stable interval (intensity value) for displaying the background. However, their algorithm cannot handle the situation where objects move at different depths. For this reason, Chia-Chih Chen et al. [10] proposed an algorithm for adaptive input sequence that can balance the inconsistency caused by different depths of objects. Unfortunately, video quality is affected by weather, terrain, jitter, and other factors, and is often less than ideal. Yuheng Deng et al. [11] restored the adaptive background to the current frame through robust spatio-temporal background initialization and by using the background information in spatio-temporal features. Guang Han et al. [12] proposed a background initialization method based on adaptive online low-rank subspace learning, which can adapt well to dynamic scenes.

Foreground detection is a key part of the field of computer vision. Its purpose is to identify the changes in the image sequence and separate the foreground image from the background image. Usually, after removing rain and snow from video data [13,14], a network will detect the significance of the compressed domain [15], and robust PCA [16] separates the foreground/background by decomposing a low-rank and sparse matrix. The use of a motion-assisted matrix restoration (MAMR) model for foreground/background separation was proposed by Xinchen Ye et al. [17]. For real-time video, Zuofeng Zhong et al. [18] combined an adaptive segmentation method with a background update strategy to extract correct foreground objects.

Video sequences must be maintained in the background in many computer vision and video analysis applications. In order to detect, track, recognize, classify, and analyze moving targets, it is vital to maintain a well-maintained background. To maintain the background, Toyama [19] developed a method called three-level processing (Wallflower). Using long observation sequences, K. Kim et al. [20] presented a codebook (CB) background subtraction algorithm. Tao Yang et al. [21] established a multi-layer background update model by analyzing the characteristics of object motion in image pixels and the results of background subtraction, and an environment with complex background maintenance was solved. A method developed by De-Zhang Peng [22] based on a clustering of Gaussian distributions with principal features was used to maintain and learn time-varying background images, which greatly reduced the cost and complexity.

In image processing, a gray image matrix is a low-rank matrix, which means that the rows and columns of the image are highly correlated, meaning that many row or column vectors are linearly correlated, and the information redundancy is high. On the one hand, the redundant information of a low-rank matrix can be used to recover the missing data. On the other hand, in deep learning, the convolution kernel has too many parameters and enormous amounts of redundant information. In this case, we can perform low-rank decomposition of the convolution kernel and decompose the convolution kernel into one and one kernel. In this way, the number of parameters can be reduced, the accuracy can be improved, and overfitting can be avoided.

In addition, the redundant information contained in high-dimensional data can obscure the potential correlations among the data. According to a large number of experimental studies, the correlation data in large amounts of high-dimensional data exist or approximately exist in a linear low-dimensional subspace. Therefore, in order to obtain the most valuable low-dimensional structure in high-dimensional data, it is vital to find an effective reduction method. Principal component analysis (PCA) [23,24,25] is a classical method that is commonly used in video or image processing. It can capture moving foreground in video well, which is consistent with our notion of capturing deformable parts in videos. However, in actual data, the observed data are often affected by large amounts of amplitude noise, so the PCA method may not show good performance in practical applications. In order to solve this problem, low-rank and sparse decomposition (LRSD) [26,27] was developed. This method can eliminate pointless redundant information in observed data and find the low-dimensional representation of the original data. It is more robust to the actual data and can satisfy more diversified practical application scenarios. Consequently, we preferred low-rank methods for improving the attention mechanisms and making them better focus on ROIs.

### 2.2. Haptic Signal Processing

Haptic communication is recognized as a promising enabler of extensive services. In the process of human–robot interaction, there is a number of applications for bilateral control in the fields of mobile robot detection and operation in extreme environments. Since vivid haptics have a short sampling period and a high frequency of communication, fast data transmission is required for their transmission. A bilateral control system is limited in its ability to be applied due to this problem. Cross-modal transmission requests are usually difficult to satisfy with existing stream schedulers because they are generally incapable of simultaneously satisfying low-latency, high-reliability, high-throughput, and low-complexity requirements. Liang Zhou et al. [28] proposed a cross-modal flow scheduling scheme for e-Health systems, which improved user experience. Ref. [29] proposed a collaborative communication mechanism for multimedia applications by exploring the temporal, spatial, and semantic correlations of cross-modal signals to strengthen the user’s immersive experience. For privacy-sensitive users in multimedia applications, Liang Zhou et al. [30] proposed a robust QoE evaluation scheme for systems with unreliable and insufficient observation data to accurately evaluate their QoE. Yan Wu et al. [31] assisted haptic communication in teleoperation systems by introducing D2D communication. Furthermore, mobile edge computing (MEC)-based computational offloading [32] allows for different latency and energy preferences for different haptic data.

This also reduces communication traffic by predicting and inferring incoming data. Because predicting multiple independent variables is difficult, traditional prediction-based methods send only one type of data (such as position, velocity, or force) in each direction. To address this issue, Satoshi Hangai et al. [33] proposed a four-channel bilateral control system based on the predicted reduction of flow, which delivered precise haptics by delivering position and force data; specifically, in an impedance control [34] structure, the control scheme was equivalently transformed, and the transmission data were aggregated into one type of data, that is, the balance force. The balance force can be expressed as follows:(1)$$f_{m\to s}^{cmd}=\frac{K_{p}+K_{v}s}{K_{f}}x_{m}-f_{m},$$
(2)$$f_{s\to m}^{cmd}=\frac{K_{p}+K_{v}s}{K_{f}}x_{s}-f_{s},$$
where $x_{m}$, $x_{s}$, $f_{m}$, and $f_{s}$ represent the position of the master, the position of the slave, the force of the master, and the force of the slave, respectively, *s* is the Laplace operator, and $K_{p}$, $K_{v}$, and $K_{f}$ are the feedback gains. By only transmitting the balance force, the amount of transmitted data is decreased, and the transmission delay is reduced. At the receiving end, the position and force can be inversely solved through inverse transformation. In addition, there is an extrapolator on the master side. As long as the extrapolation force differs from the ground truth by less than the error threshold, there is no need to transmit the force value, and the extrapolation force is used on the slave side, which further reduces the amount of data transmitted.

### 2.3. Video Reconstruction of Haptic Information

In recent years, more and more studies have utilized convolutional neural networks (CNNs) [35] to process images. As a result of their performance in various sequence-based applications, recurrent neural networks (RNNs) have achieved good performance, but the problem of gradient vanishing and explosion remains unsolved, which brings difficulties for long-term dynamic learning. To this end, Sepp Hochreiter et al. [36] proposed long short-term memory (LSTM). Wonjun Hwang et al. [37] used LSTM and CNN networks to estimate the contact force by only relying on images without using sensors. To make the results more accurate, they added electrical signals [38]. Dongyi Kim et al. [39] proved that 3D-CNN has higher accuracy than 2D-CNN. Consecutive image frames [40] are beneficial for the network to estimate the output more accurately through the preceding and following frames. Hochul Shin et al. also added an attention mechanism [41] to the neural network using bidirectional LSTM (BLSTM) [42], which experimentally demonstrated that the attention mechanism could improve the accuracy of the neural network.

Foreground separation and the development of CNNs have made it possible to separate the foreground from video and to extract foreground features. Consequently, we can use video information and a CNN to reconstruct tactile information and reduce the tactile data required to be transmitted.

## 3. Proposed Method

Our network extracts features from consecutive video frames through a CNN, and then infers the predicted force. Unlike in a major previous work, our neural network utilizes GOPs and relies exclusively on a CNN for contact force estimation. We also incorporate attention mechanisms into the neural network to increase the estimation accuracy. The overall architecture of the LRCNN proposed in this paper is shown in Figure 1; it includes GOP preprocessing and a CNN with attention mechanisms. Initially, continuous video frames are processed, and then we extract features in the LRCNN. This step reduces the computational complexity and extracts important features with higher-level discrimination. Then, the LRCNN uses the visual features to make predictions. The purpose of this section is to individually introduce the preprocessing, attention mechanisms, and LRCNN that we used.

### 3.1. Preprocessing

We define *n* video frames as $X=\{x_{1},x_{2},\cdots ,x_{t},\cdots ,x_{n}\}$, where $x_{t}$ represents the video frame at time *t*, and each video frame in *X* corresponds to a force value $f_{t}$. For continuous video frames, we believe that there is a relationship for estimating force in certain video frames. Here, the tactile information of the current video frame can be reconstructed with the video frames. Ref. [43] took multiple related scenes in a house as input, let the network adaptively find the correlation, and then predicted the house price. In a similar way, for a video stream, as we all know, a video can be divided into many GOPs, which contain *N* frames with an IPPP structure. Taking a GOP as an example, the IPPP structure starts with an intra-coded *I* frame and is followed by some predicted frames, which we define as *P* frames. In the *I* frame, it encodes by itself and does not rely on any other frames. However, in a *P* frame, it encodes by preceding based on an *I* frame. Therefore, we realize that the *I* frame in a GOP has a greater significance than the *P* frames, since the later *P* frames decode depending on the previous *I* frame. Based on this idea, we splice GOPs to obtain the input $C=\{c_{1},c_{2},\cdots ,c_{n/4}\}$, $\left\{x_{t},x_{\left(t+1\right)},x_{\left(t+2\right)},x_{\left(t+3\right)}\right\}\in c_{t/4}$, and the corresponding force value label is ${\bar{f}}_{t/4}=\frac{f_{t}+f_{\left(t+1\right)}+f_{\left(t+2\right)}+f_{\left(t+3\right)}}{4}$, so that the network can extract features from the frames before and after the current video frame. The output is the force value ${\hat{f}}_{avg}$. The preprocessing of continuous video frames is shown in Figure 2.

As a result, it is important to keep in mind that the inter-frame gap of consecutive video frames should be tiny enough. Hence, the difference between the force values corresponding to adjacent images is small enough so that the error of the average force value will not be too obvious. Here, we assume that the difference between the predicted force value and the actual value is less than the just noticeable distortion (JND), so the error is negligible.

### 3.2. LAM

During feature extraction, we want the LRCNN to pay more attention to the parts that we are interested in and pay less attention to the parts in which we have less interest. During the contact process, the area where the deformation occurs at the contact position is the area that should be paid more attention to. It is not possible for us to manually modify where the LRCNN should pay attention. Therefore, we add an attention mechanism [44,45,46] in the downsampling. The spatial attention mechanism, channel attention mechanism, and the combination of the two are the three general types of attention mechanisms. This chapter will introduce the proposed low-rank foreground-based spatial attention mechanism (LSAM), low-rank foreground-based channel attention mechanism (LCAM), and their combination, namely, LAM. The attention mechanisms take the extracted features as input and output features with weights. As the network trains, the network adaptively assigns larger weights to the places of interest, and vice versa. Below, we depict the attention mechanisms that we apply in Figure 3 and Figure 4.

Generally speaking, the contact force will occur in the contact position of the object. For a deformed object, the magnitude of the force is related to the degree of deformation. Except for the region of the contact position, it is hard to provide useful information for reconstructing the haptic signal, which even causes interference and leads to sub-optimal solutions. To solve this problem, inspired by [47], we propose an LSAM. This mechanism can allocate more weight to the foreground region by separating the low-rank foreground and sparse background of the video frame so that the CNN can find the parts that need attention in the foreground faster and better. The weight distribution process is illustrated in the box in Figure 3. Here, we assume that, in an ideal situation, the weight value of the concerned part is 1, and the weight value of the uninteresting part is completely ignored, so there is no need to assign a weight, or it is 0. In Figure 3, we denote the convolutional feature of the image as $F\in R^{W\times H\times C}$. In addition, we perform global average pooling on the feature to obtain $F^{\prime }\in R^{W\times H\times 1}$. Then, we assign a weight matrix $M_{s}\in R^{W\times H}$ for each pixel through the sigmoid activation function with a value of 0∼1, which is denoted as
(3)$$M_{s}=\sigma \left(F^{\prime }\right),$$
where σ is the sigmoid function. $x_{i,j}\in M_{s}$ represents the linear combination of all channels in the space (i,j). Therefore, the spatial attention mechanism can be expressed as
(4)$$\tilde{F}=M_{s}⊗F,$$
where ⊗ represents the corresponding element multiplication operation, and $\tilde{F}\in R^{W\times H\times C}$ represents the feature when the weight matrix is allocated. The weight in the contact position and the deformation are larger than those in other positions, as shown in the red area on the right side of Figure 3. Through iteration, the CNN adaptively allocates more weight to the length and width dimensions of the region of interest in the foreground and reduces the weight of unimportant background regions so as to improve performance. Assuming that the video frame is 4×4 pixels, let the weight matrix of the feature obtained by the attention mechanism be
(5)$$H_{f}=\left[\begin{array}{cccc} 0 & 0 & 1 & 0\\ 0 & 0 & 1 & 0\\ 1 & 1 & 1 & 1\\ 0 & 0 & 0 & 0 \end{array}\right],$$
where 1 is the weight of the foreground position in the video frame, and 0 corresponds to the background position. Through iteration, the CNN adaptively updates the weight:(6)$$H_{f}=\left[\begin{array}{cccc} 0 & 0 & 0.4 & 0\\ 0 & 0 & 1 & 0\\ 0.4 & 1 & 1 & 0.4\\ 0 & 0 & 0 & 0 \end{array}\right],$$
where 1 corresponds to the weight of the deformation position, and 0.4 corresponds to the weight of other foreground positions.

Similarly to the spatial attention mechanism, a channel attention mechanism [48] is implemented by assigning weights to channels. We propose an LCAM. How it works is shown in Figure 4. For RGB video frames, this mechanism assigns a greater weight to the color of objects in the low-rank foreground, such as the red of a sponge. $F^{\prime }\in R^{1\times 1\times C}$ is obtained after input features $F\in R^{W\times H\times C}$ go through the full connection layer, and weights are assigned to each channel through the sigmoid activation function, which is regarded as
(7)$$M_{c}=\sigma \left(F^{\prime }\right),$$
where $M_{c}\in R^{1\times 1\times C}$. The channel attention mechanism in our scheme can be expressed as:(8)$$\tilde{F}=M_{c}⊗F,$$
where $\tilde{F}\in R^{W\times H\times C}$.

We also combine the LSAM and LCAM to compare and contrast with the performance of each mechanism alone. This structure is shown in Figure 5.

### 3.3. LRCNN

For the part of the video frame in *X*, our feature extraction process is completed with the help of a neural network. Therefore, we propose an LRCNN to handle this issue. In addition to improving the structure of the VGG16 network [49], it also contains convolutional layers, pooling layers, fully connected layers, and layers with activation functions. The reason that we chose a CNN is because in artificial neural networks (ANNs), with the increase in image size, the number of trainable parameters increases sharply. In order to process data, the first problem to be resolved is that of tensoring the data. The data to be processed by an ANN must be in vector form. For the data type of images, if they are expanded into a one-dimensional vector, in addition to getting the vector, if the dimension is too high, the network will be too deep, resulting in there being too many parameters in the network, and the spatial information in the image will also be lost. A CNN can extract “partial specific information” from the original information through convolution, and it is natively supported for two-dimensional images, thus preserving the spatial information in the image. The parameters of the convolutional neural network are only the parameters of the convolution kernel and biases. The parameters of the convolution kernel can be shared, and the convolution kernel can also be used to interpret the original image from multiple angles. In addition, a CNN is more mature than the ANFIS [50] proposed in 1993. Fuzzy logic has some applications in natural language processing, but linear and nonlinear processes used in CNNs and RNNs can often better replace them. The specific structure of the network is shown in Table 1. Below, we describe the procedure of feature extraction in detail.

Convolutional layers convolve the input features with the filter bank to generate output features. The equation of the *k*-th convolutional layer can be expressed as follows:(9)$$z_{p}^{k}\left(i,j\right)=\sum_{q=1}^{n^{k-1}}\sum_{u=-1}^{1}\sum_{v=-1}^{1}\alpha_{q}^{k-1}\left(i-u\right)l_{p,q}^{k}\left(u,v\right)+b_{p}^{k},$$
where
(10)αqk(i,j)=ReLU(zpk(i,j)),
where $n^{k}$ is the number of convolution kernels in the *k*-th layer, $l_{p,q}^{k}$ is the convolution kernel corresponding to the *p*-channel of the *k*-th layer and the *q*-channel of the *k*-1 layer, $b_{p}^{k}$ is the offset of the *p*-channel of the *k*-th layer, $z^{k}$ is the *k*-th layer without an activation function, $\alpha^{k}$ is the output of the *k*-th layer after the activation function, and ReLU(•) is the activation function.

In a CNN, a pooling layer is usually added to connect adjacent convolutional layers. The pooling layer can effectively reduce the size of the parameter matrix, thereby reducing the number of parameters in the final fully connected layer. Therefore, adding a pooling layer can speed up the computation and prevent overfitting. There is a maximum pooling layer and an average pooling layer in VGG16, and the maximum pooling formula of the *k*-th layer is:(11)$$z_{p}^{k}\left(i,j\right)=\max\left(\alpha_{p}^{k-1}\left(2i-u,2j-v\right)\right),$$
where u,v∈{0,1}. The average pooling formula for the *k*-th layer is as follows:(12)$$z_{p}^{k}\left(i,j\right)=avg\left(\alpha_{p}^{k}\left(7-u,7-v\right)\right),$$
where u,v∈{0,1,2,3,4,5,6}.

We use sleep function dropout after each pooling layer, set it to β, and cause the neurons to sleep with a proportion of β to avoid overfitting. Before transmitting the extracted features to the fully connected layer, we send them to a “flatten” layer, which pulls the pooled data into a one-dimensional vector to represent them and is convenient for input into the fully connected network.

The first two layers of the fully connected layer use dropout, which is set to *d*. With $r^{k}$, we can represent the connectivity between nodes in the *k*-th layer, which follows the Bernoulli distribution:(13)$$r^{k}∼Bernoulli\left(d\right).$$

Then, the output of the *k*-th fully connected layer is
(14)zpk=ReLU(Wpk(rp×αpk)+bpk),
where $W^{k}$ represents the weight of the fully connected network in the *k*-th layer. Through the fully connected layer, we can get the feature map $X^{F}=\left\{x_{1}^{F},x_{2}^{F},\cdots ,x_{t}^{F},\cdots ,x_{n}^{F}\right\}$. In the traditional VGG16 network, in the last layer with fully connected neurons, there is the same number of labels as neurons. Finally, through the activation function SoftMax(•), the outputs of multiple neurons are equally mapped to the interval (0,1), and the probability of the predicted value $p_{n}^{x}$ can be expressed as follows:(15)$$p_{n}^{x}=\frac{e^{x_{i}^{F}}}{\sum_{j=1}^{n}e^{x_{j}^{F}}},$$
where
(16)$$\sum_{j=1}^{n}e^{x_{j}^{F}}=1.$$

The role of the fully connected layer is that of a classifier, which maps the extracted features to labels. Because our final output consists of one value, we set the number of neurons in the last fully connected layer to one and use the activation function Linear(•), whose structure is shown in the dotted box in Figure 6.

We use mean squared error (MSE) as the loss function as follows:(17)$$L\left(\hat{y},y\right)=\frac{1}{n}\sum_{i=1}^{n}{\left(y_{i}-{\hat{y}}_{i}\right)}^{2},$$

In this example, $\hat{y}$ represents the predicted interaction force, while *y* represents the ground truth.

## 4. Experiment and Evaluation

### 4.1. Dataset and Preprocessing

A contact force database [40] was used to test the performance of our method in this experiment. As shown in Figure 7, this dataset contained contact videos of four objects (paper cup, sponge, stapler, and tube). The video content was touched from four different angles ($0^{\circ }$, $10^{\circ }$, $20^{\circ }$, $30^{\circ }$) and three illuminations (350, 550, and 750 lux). The total number of units of data was 17×4×4×3 video segments, and there were a total of 816 video clips, each of which contained a complete process of four touches. Each clip contained about 500 consecutive video frames. The total number of images was 440,967, and each image was 128 × 128 (RGB) pixels. Each video frame corresponded to a force value measured by a sensor. So, there were 816 text files recording tactile data. In addition, the dataset was obtained by applying different forces to the object with the manipulator at the same speed. If the force applied by the robotic arm was too rapid, the force difference between successive frames could become larger, the difference between the average force value between GOPs and the current frame force value would rise, and the accuracy would decrease. We could solve this problem by increasing the frequency of the camera and decreasing the interval between frames. The data collection device was composed of an RC servo-motor and a probe. If the probe would be too small, it would not be conducive to the attention mechanism for selecting ROIs, and the probe could be excluded as noise. If the probe would be too large, the deformation of the contact position could occur, and the predicted value would be less than the ground truth. For elastic deformable objects of a single material, such as the sponge and paper cup, applying force at different positions on the surface can produce evidently deformable objects and will not cause difficulty in feature extraction because of the different positions of the contact points. Nevertheless, for inelastic deformation of objects composed of multiple materials, such as the stapler and tube, the deformation generated by applying force at different positions will be different according to the different materials, and the different positions of contact points may affect the feature extraction and decrease the performance. All of the datasets used applied force at the center of the object and measured force values only in the *z* direction. Since the video frame rate was 120 fps, we supposed that the force difference between the first and fourth frames was less than the JND. We averaged the force values of four consecutive video frames as a label force value in this GOP. We performed the experiments with different materials when we trained all illuminations and angles of the video frames in the same material together. We took 20% of the video frames in the dataset as the validation set, and there was no intersection between the training set and the validation set.

### 4.2. Experimental Setup

The hyperparameters of the LRCNN in the experiment are shown in Table 2.

Our experiments were based on the TensorFlow framework [51], using computers with the following specific configurations: AMD EPYC 7543 32-core processor CPU and RTX 3090 × 2 GPU. During the training, the Adam optimizer was used to update the network parameters and minimize the MSE loss function. Initially, lr=0.001, and the learning rate was reduced with each iteration. After *i* iterations, the learning rate was
(18)$$lr=lr\times \frac{1}{1+\alpha \times i},$$
where α is the decay and the epoch is the number of network iterations. We added dropout after each convolutional layer to make 25% of the neurons hibernate to avoid overfitting.

### 4.3. Results and Discussion

First, we took a single video frame as the input of the network to test its performance, and the result is shown in Figure 8.

As can be seen from Figure 8, even for the paper cup and sponge, which had extremely obvious deformation, there was still a big difference between the prediction results obtained by the network with a single frame as input and the ground truth.

We then tested the performance of the LRCNN with GOPs as input; Figure 9 shows the results.

The following can be seen from Figure 9: The LRCNN with GOPs as input was more accurate than that with single video frames, again proving that there was indeed a correlation between frames within the GOPs. This method can obviously improve the performance with nonlinearly elastically deformable objects, such as a stapler and a tube. In particular, when estimating the contact force on the stapler, the training loss of the method with a single video frame as the input continued to decline after 20 iterations, but the verification loss began to rise, as shown in Figure 10, indicating that overfitting occurred. This phenomenon did not occur with the LRCNN with GOPs as input because, by using GOPs as input, the network could extract more features and avoid overfitting to some extent. The validation loss of the two methods is shown in Table 3.

Figure 11 shows the performance results when adding the LAM to the LRCNN. For the paper cup and sponge, the performance was improved, especially for negative force values (due to the measurement error in the dataset [40], the individual force values were negative when there was no contact, so the predicted force values were negative), which were closer to 0. When the three networks predicted the contact force, the peak force was always biased, and the reasons were as follows: (1) The degree of deformation of the object was too large, resulting in partial occlusion of the deformation, the feature could not be extracted, and the predicted force was higher than the ground truth (such as with a sponge). (2) When the deformation of the object reached the limit, the object that continued to apply force would not continue to be deformed, or the object that needed more force would continue to deform. In this case, the predicted force would be lower than the ground truth (as with the stapler).

Table 3 illustrates the MSE for force prediction for the four materials with the three networks, and we again confirmed that the proposed method that used GOPs as input improved the performance in most force intervals. Although the predicted force could not completely fit the ground truth, a practical application could ignore the difference between the predicted force and ground truth, since it is smaller than the JND. In addition, the network structure of our method is simpler, and a compromise between network complexity and performance is considered to achieve optimal results.

In order to verify the accuracy of our method, we took the peak value of the force in the experiment for comparison. We first calculated the deviation between the ground truth of each peak and the predicted value as follows:(19)$$a=\frac{\left|y_{\max}-{\hat{y}}_{\max}\right|}{y_{\max}},$$
where, $y_{\max}$ is the maximum force value of the ground truth with a touch, and ${\hat{y}}_{\max}$ is the corresponding predictive force value. After calculating all of the deviations from a validation, we averaged the deviations from the prediction of the maximum force value:(20)$$\bar{a}=\frac{\sum_{i=1}^{n}a_{i}}{n},$$
where *n* is the number of touches. Each of our validations included 16 touches, so *n* = 16. For the LRCNN, for the four materials, the predicted deviations of the maximum force value were 23.97%, 47.75%, 17.39%, and 15.58%, respectively. For the LRCNN + LAM, the predicted deviations were 19.48%, 9.42%, 9.61%, and 13.87%, respectively. It can be seen that, after adding LAM, the prediction error was smaller than that of the method in which LAM was not added.

We also separately added LSAM to the network to compare the performance with that of the other networks. The results are shown in Table 4. It can be seen that, for the sponge and stapler, the performance of our method was improved by 96.2% and 61.9% compared to that of the LRCNN, as shown in Figure 12. When one attention mechanism is used alone, the network performance may be better than when two attention mechanisms are used simultaneously because, under the conditions of mixed lighting data, multiple types of lighting lead to an increase in the parameters of the channel attention mechanism, which increases the probability of overfitting.

To validate our idea, we re-ran the above experiments using the single-light-condition dataset, and the results are shown in Figure 13.

In Figure 13, it can be seen that the network with the addition of the dual attention mechanism outperformed the network with only a single attention mechanism. For the four materials, our method improved the performance by 31.7%, 38.0%, 88.2%, and 54.2%, respectively, compared to that of the LRCNN. Our method improved the performance with inelastically deformable objects more significantly. This was because elastically deformable objects were significantly deformed and there were sufficient features to extract. Therefore, the performance of the LRCNN was close to saturation, and the deformation of the inelastically deformed objects was not obvious. With our method, it is possible to better focus on the smallest changes and extract the features in more concentrated areas.

Due to limitations of the dataset, all video frames employed were shot from the same direction. For deformable objects, such as sponges and paper cups, obvious deformations can be observed when photographing from different angles, and a CNN can perform feature extraction well. For two types of inelastically deformable objects—a stapler and a tube—however, the degree of deformation observed could be different when taken from different angles. For example, a tube is concave in the middle and protrudes on both sides. If the object is shot from the tail, the deformation of the object can be better observed, which is conducive for the attention mechanism’s assignment of a more reasonable weight to the ROI and for the CNN to better extract features for prediction. However, with the same shooting angle, our proposed method can extract the features of the deformed parts better than the method that does not use the attention mechanisms.

## 5. Conclusions

We proposed an LRCNN with LAM to predict contact force. The LRCNN was constructed with VGG16 as the framework, and then LAM was added to extract the visual features of GOPs and, ultimately, predict the contact force. The method proposed in this paper has made a great contribution to HRI. On the one hand, the network can sufficiently exploit the correlation between GOPs. On the other hand, we incorporated attention mechanisms that could better extract visual features into the network, and the predicted contact force was closer to the ground truth. The experiments showed that the network had better prediction performance while reducing network complexity compared to the LRCNN, which took a single frame as input and did not use an attention mechanism. After eliminating the effects of various illuminations, our method improved the performance of the LRCNN by more than 30%. In the complex case of mixed lighting, our proposed method also outperformed the LRCNN in most force intervals, especially for objects with obvious deformations.

We believe that our research can facilitate tactile signal restoration in HRI, making the conditions required for precise tactile signals easier. In our plan, we will add a material recognition function on the basis of the existing network and will predict the contact force based on this material recognition so as to make the results more accurate. We will also look for more suitable datasets for the verification of our experiments, and when we have enough data, we will try to obtain the distribution law of contact force through fitting, and then show it through simulation.

## Figures and Tables

**Figure 1 sensors-22-06517-f001:**
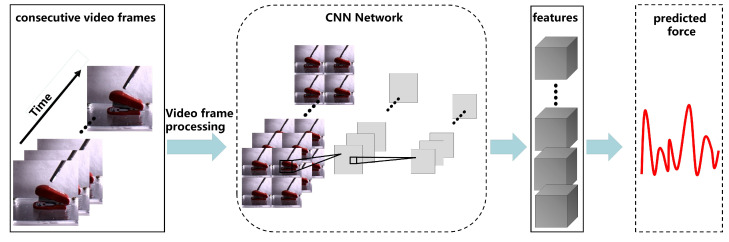
Overall architecture of the LRCNN.

**Figure 2 sensors-22-06517-f002:**
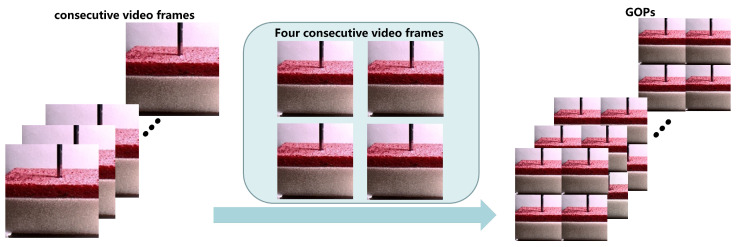
Preprocessing of video frames.

**Figure 3 sensors-22-06517-f003:**
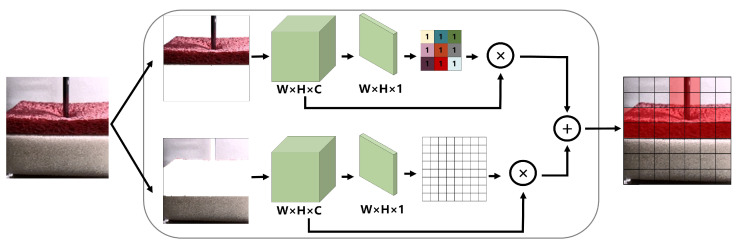
Low-rank foreground-based spatial attention mechanism.

**Figure 4 sensors-22-06517-f004:**
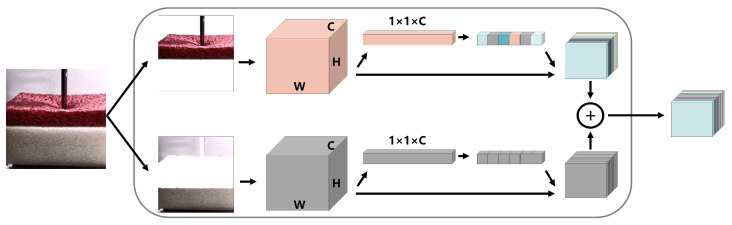
Low-rank foreground-based channel attention mechanism.

**Figure 5 sensors-22-06517-f005:**
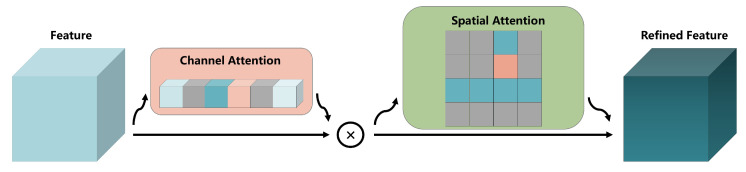
Architecture of the combined network.

**Figure 6 sensors-22-06517-f006:**
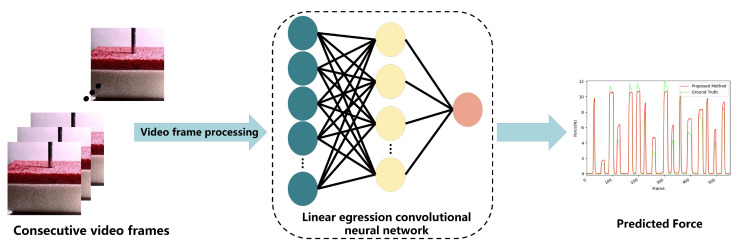
The network structure of the LRCNN.

**Figure 7 sensors-22-06517-f007:**
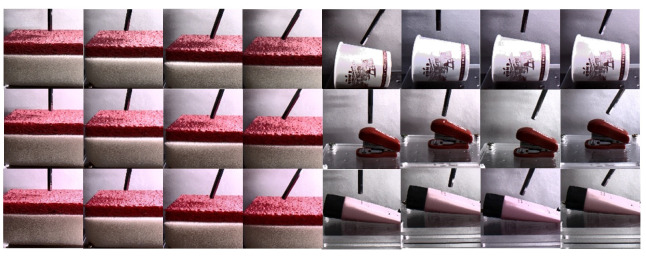
Dataset collected to estimate interaction forces.

**Figure 8 sensors-22-06517-f008:**
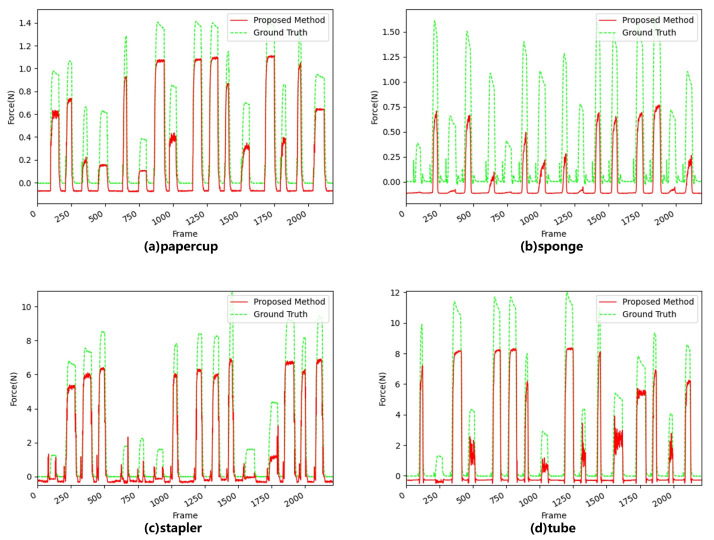
Performance of a single frame as input. Where green is the ground truth and red is the predictive force value.

**Figure 9 sensors-22-06517-f009:**
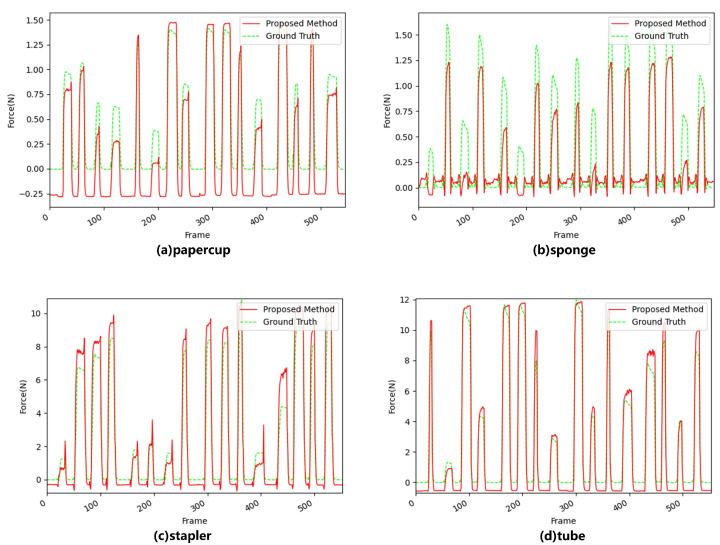
Performance results of the LRCNN, where red is the predicted value of the proposed method and green is the ground truth.

**Figure 10 sensors-22-06517-f010:**
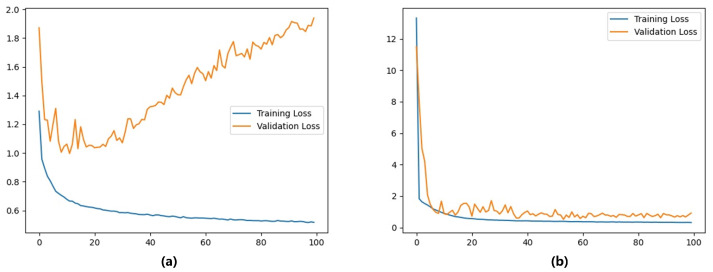
Loss when estimating the stapler contact force for two networks: (**a**) single video frame as input and (**b**) LRCNN.

**Figure 11 sensors-22-06517-f011:**
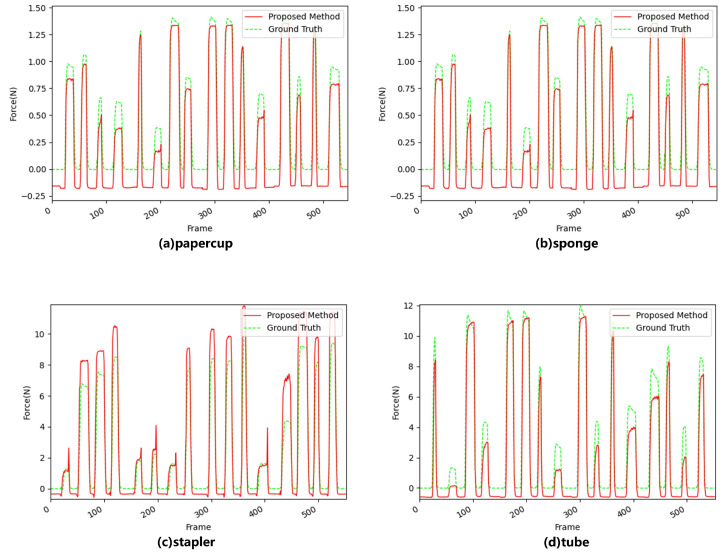
Predicted value and ground truth value results of the estimation of the interaction force using the LRCNN with LAM.

**Figure 12 sensors-22-06517-f012:**
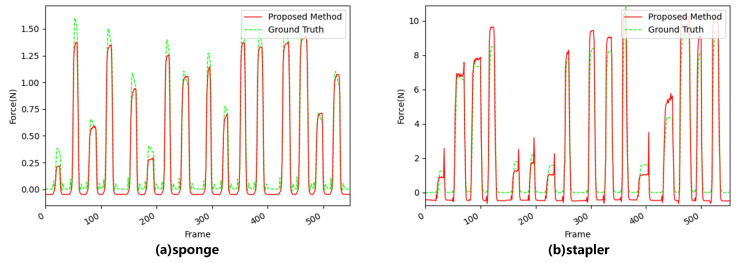
Performance when using LSAM separately for (**a**) the sponge and (**b**) the stapler.

**Figure 13 sensors-22-06517-f013:**
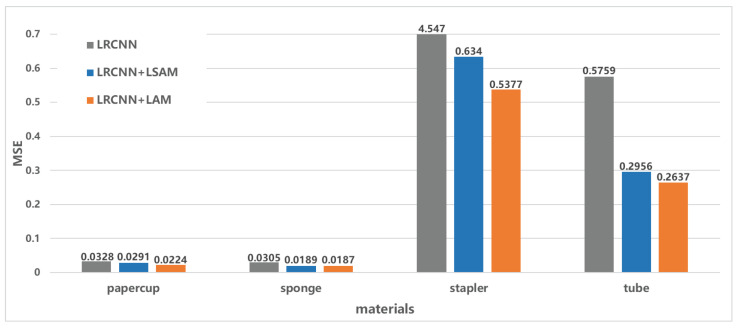
Comparison of the performance of the three networks with a single source of illumination.

**Table 1 sensors-22-06517-t001:** Network architecture.

Layers	Types	Kernel Nums
Layer1-1	3×3 conv	64
Layer1-2	3×3 conv	64
Pooling1	Maxpooling (stride2)	
Layer2-1	3×3 conv	128
Layer2-2	3×3 conv	128
Pooling2	Maxpooling (stride2)	
Layer3-1	3×3 conv	256
Layer3-2	3×3 conv	256
Layer3-3	3×3 conv	256
Pooling3	Maxpooling (stride2)	
Layer4-1	3×3 conv	512
Layer4-2	3×3 conv	512
Layer4-2	3×3 conv	512
Pooling4	Maxpooling (stride2)	
Layer5-1	3×3 conv	512
Layer5-2	3×3 conv	512
Layer5-2	3×3 conv	512
Pooling2	Maxpooling (stride2)	
Flatten	Flatten ()	
Fully connect 1	FC-4096	
Fully connect 2	FC-4096	
Fully connect 3	FC-1	

**Table 2 sensors-22-06517-t002:** Hyperparameters for the LRCNN.

	lr	α	Epoch	Input Size	Batch Size
LRCNN	0.001	0.01	100	128 × 128 × 3	64

**Table 3 sensors-22-06517-t003:** Validation loss of the three methods. The best results are in bold.

	Paper Cup	Sponge	Stapler	Tube
single input	0.0543	0.2448	1.9392	2.4693
LRCNN	0.0654	0.0643	**0.9133**	**0.7434**
LRCNN+LAM	**0.0300**	**0.0256**	1.0560	0.8656

The best result is in bold.

**Table 4 sensors-22-06517-t004:** Validation loss of the LRCNN with LSAM.

	Papercup	Sponge	Stapler	Tube
LRCNN + LSAM	0.0440	**0.0092**	**0.7375**	0.8566

The best result is in bold.

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
