# Peer review of "Cross-Modal Reconstruction for Tactile Signal in Human–Robot Interaction"

_sensors, 2022, doi:10.3390/s22176517_

Round 1

Reviewer 1 Report

This paper presents an algorithm to decode force information from video frames that contain interactions with the object. The algorithm consists of general CNN network and attention mechanism to spatially discriminate the point of interest. It is of great interest as the fields of application in human-robot interactions. Here are some minor comments:

1)      How will the prediction vary by the speed of the force and size of the robotic arm?

2)      The selection of the window (about 500 frames) was not justified strongly in the paper. The work could benefit from showing the performance when considering different window sizes.

3)      Could authors include accuracy of the system?

4)      Authors may comment how the system can deal with varying conditions such as camera angles.

Reviewer 2 Report

Dear authors,

Have a look my comments

Round 2

Reviewer 2 Report

Done